# Teaching a new mouse old tricks: Humanized mice as an infection model for *Variola virus*

Christina L. Hutson[1]☯*, Ashley V. Kondas[1]☯, Jana M. Ritter[2], Zachary Reed[1], Sharon Dietz Ostergaard[3], Clint N. Morgan[1], Nadia Gallardo-Romero[1], Cassandra Tansey[3], Matthew R. Mauldin[1], Johanna S. Salzer[1], Christine M. Hughes[1], Cynthia S. Goldsmith[2], Darin Carroll[1], Victoria A. Olson[1]

**1** Poxvirus and Rabies Branch, Centers for Disease Control and Prevention, Atlanta, Georgia, United States of America, **2** Infectious Diseases Pathology Branch, Centers for Disease Control and Prevention, Atlanta, Georgia, United States of America, **3** Comparative Medicine Branch, Centers for Disease Control and Prevention, Atlanta, Georgia, United States of America

☯ These authors contributed equally to this work.

\* zuu6@cdc.gov

**Data Availability Statement:** All relevant data are within the manuscript and its Supporting Information files.

## Abstract

Smallpox, caused by the solely human pathogen Variola virus (VARV), was declared eradicated in 1980. While known VARV stocks are secure, smallpox remains a bioterrorist threat agent. Recent U.S. Food and Drug Administration approval of the first smallpox anti-viral (tecovirimat) therapeutic was a successful step forward in smallpox preparedness; however, orthopoxviruses can become resistant to treatment, suggesting a multi-therapeutic approach is necessary. Animal models are required for testing medical countermeasures (MCMs) and ideally MCMs are tested directly against the pathogen of interest. Since VARV only infects humans, a representative animal model for testing therapeutics directly against VARV remains a challenge. Here we show that three different humanized mice strains are highly susceptible to VARV infection, establishing the first small animal model using VARV. In comparison, the non-humanized, immunosuppressed background mouse was not susceptible to systemic VARV infection. Following an intranasal VARV challenge that mimics the natural route for human smallpox transmission, the virus spread systemically within the humanized mouse before mortality (~ 13 days post infection), similar to the time from exposure to symptom onset for ordinary human smallpox. Our identification of a permissive/representative VARV animal model can facilitate testing of MCMs in a manner consistent with their intended use.

## Author summary

Preparedness activities against highly transmissible respiratory viruses with high mortality have been highlighted during the ongoing COVID-19 pandemic. Smallpox, caused by Variola virus (VARV) infection, is highly transmissible and estimated to have killed 300 million people in the 20th century alone with ~30% mortality. Through an intensive vaccination campaign smallpox was declared eradicated in 1980 and routine smallpox vaccination of individuals ceased. Today's population has little/no immunity against VARV; if

**Funding:** This study was funded by Biomedical Advanced Research and Development Authority (VO). The funders had no role in study design, data collection and analysis, decision to publish, or preparation of the manuscript."

**Competing interests:** The authors have declared that no competing interests exist.

smallpox were to re-emerge the worldwide results would be devastating. Development of antiviral strategies is critical for outbreak response efforts and in order to truly gauge the potential effectiveness of a medical countermeasure (MCM), it must be tested directly against the pathogen. VARV is solely a human pathogen and infections of surrogate animal models have been largely unsuccessful. Here we use the humanized mouse to establish the first small animal model using VARV. We found that the virus spread systemically within humanized mice before mortality (~ 13 days post infection), similar to the time from exposure to symptom onset for ordinary human smallpox. These findings improve upon current MCM efficacy testing methods by providing a means to conduct *in vivo* evaluations of smallpox MCMs directly against VARV.

## Introduction

Variola virus, the causative agent of smallpox, is a solely human pathogen. The World Health Organization (WHO) certified the global eradication of smallpox in 1980 with the last reported naturally occurring case in Somalia in 1977. While the known viral stocks are secure in two laboratories (U.S. Centers for Disease Control and Prevention [CDC] and VECTOR Institute), the threat of unknown sources outside the repositories remains and could be used with malicious intent. *Variola virus* (VARV) is regulated by the U.S. Federal Select Agent Program, which oversees the possession, use and transfer of biological select agents and toxins, which have the potential to pose a severe threat to public. Forgotten lyophilized VARV vials from the 1950's were discovered in 2014 at a non-WHO approved variola virus laboratory [1]. Additionally, the procedure for recreating an OPXV has been documented (*Horsepox virus*) thereby providing means to recreate a human pathogenic OPXV such as VARV [2]. A recent tabletop exercise evaluating what would happen if smallpox was used as a bioweapon indicated potentially devastating consequences [3]. While two vaccines (*Vaccinia virus*; ACAM 2000 and JYN-NEOS) and one anti-viral (TPOXX [ST-246]) have received licensure by the U.S. Food and Drug Administration (FDA) as medical countermeasures (MCMs) for use against smallpox infection, these MCMs were developed after the eradication of human smallpox and have never been tested against the authentic agent in humans. Because VARV causes smallpox only in humans, and thus far no satisfactory VARV animal model has been developed, the U.S. FDA has approved The Animal Model Rule as a pathway for regulatory approval. Under this rule, potential therapeutics must be tested in at least two surrogate *Orthopoxvirus* animal models, such as *Monkeypox virus* in non-human primates (NHP), *Rabbitpox virus* in rabbits and/or *Ectromelia virus* (ECTV) in mice (21 CFR 601.90). While this route to U.S. FDA licensure is critical for preparedness, the development of a small animal model susceptible to VARV infection would be ideal for testing MCMs directly against the authentic agent.

Humanized mice have become a valuable tool for studying infectious diseases [4–6]. An animal model with a human-like immune system, would be advantageous to the study of VARV, potentially identifying why VARV is a solely human pathogen and providing valuable samples for understanding aspects of the human immune response to VARV infection. Given this, we sought to determine if humanized mice can support a productive VARV infection. Three different strains of humanized mice (hu-PBMC, hu-CD34+, and hu-BLT) were evaluated and compared to the susceptibility of the immunodeficient NSG background mouse. This sequence of studies allowed us to better understand if susceptibility to VARV infection is attributable to a lack of murine immune response, the humanization process or some combination of both. To produce these mice, the NOD scid gamma (NSG) background mouse

undergoes different methods of humanization. For humanized-PBMC mice (hu-PBMC), the NSG mouse is engrafted with human peripheral blood mononuclear cells (PBMC) which produces a strong human T cell engraftment (effector and memory T cells). Humanized-CD34+ (hu-CD34+) mice are created by engrafting the NSG mouse with human hematopoietic stem cells (CD34+ cells) which produces circulating human CD4$^+$ and CD8$^+$ T cells. Humanized-BLT (hu-BLT) mice, which are considered to be the most humanized because they develop human T and B cells, are created by both engrafting CD34+ cells and human fetal liver and thymus under the murine renal capsule. We found that all three strains of humanized mice were susceptible to systemic VARV infection following intranasal (IN) inoculation, while NSG background mice were not. Herein, we present novel humanized mouse models of smallpox that utilize the exact etiologic agent of human disease and provide further evidence that VARV dissemination requires some component of the human immune system.

## Results

### VARV disease in humanized mice

For part one of the study, humanized mice were challenged intranasally with one of the following: a high dose of VARV, a low dose of VARV, diluent for a non-infectious control, ɣ-irradiated VARV or received no challenge material (Table 1 and Fig 1A). The first clinical signs observed were suspect pox lesions on the hocks of a subset of mice beginning at 7 days post infection (dpi). Swabs of the lesions were collected. The hock lesions were sporadic, with only five infected animals affected, but did include animals of all three strains and no uninfected control animals (Table 1). For hu-CD34+, and hu-BLT infected groups, onset of other clinical signs was variable but generally began between 7–17 dpi (hu-CD34+ high dose), 17–19 dpi (hu-CD34+ low dose), and 12–14 dpi (hu-BLT both doses). The majority of hu-PBMC infected mice did not show clinical signs attributed to viral infection, with only three animals displaying clinical signs close to study end (17–21 dpi for both high (1/4) and low-dose groups (2/4)). The main clinical sign was >20% weight loss. Clinical disease progressed rapidly for hu-CD34 + and hu-BLT and animals were not always able to be euthanized before succumbing to disease. Hu-CD34+ 8 (low dose group) was believed to have an unsuccessful inoculation. While necropsy samples from this animal had very low levels of viral DNA in the spleen, kidney and ovaries (46.0 fg/µl, 50.5 fg/µl and 43.8 fg/µl respectively), this animal displayed no clinical signs and survived to study end. Unsuccessful inoculation was further supported by lack of pathologic lesions, viral immunostaining, and absence of detectable viable virus. Upon review of the clinical record, this mouse experienced a "bubble" during the inoculation supporting the hypothesis that the animal was not successfully inoculated (i.e., entire inoculum not delivered into nares). This animal was excluded from the study.

Dose-dependent mortality was seen in the hu-CD34+ and hu-BLT mice beginning at day 13 (Fig 1B and 1C). High dose hu-CD34+ mice were significantly more likely to succumb earlier than those in the low dose group (p = 0.02). While differences in survivorship between dose groups for hu-BLT mice were not significant (p = 0.2), there was a trend suggesting the high dose group succumbed earlier than those challenged with the low dose. All hu-BLT and hu-CD34+ mice in the high dose group and low dose group succumbed to disease. The hu-PBMC mice did not have high morbidity, and only one from the high dose group was euthanized before study end (19 dpi). Hu-PBMC mice were significantly more likely to survive VARV infection compared to hu-BLT and hu-CD34+, controlling for dose (high dose p = 0.004, low dose p = 0.008). There was no significant difference in survivorship between hu-BLT and hu-CD34+ mice when controlling for dose (high dose p = 0.20; low dose = 0.63). One hu-PBMC control mouse (challenged with ɣ-irradiated VARV) was found deceased on day 13,

**Table 1. Clinical signs and gross findings.** Humanized mice (hu–BLT, hu–CD34+ and hu–PBMC) were inoculated via the intranasal route with $7 \times 10^3$ or $7 \times 10^5$ plaque forming units (pfu) of VARV (VARV_JAP51_hrpr (primary clade I)) (n = 4 per group). Clinical signs were recorded daily, and weights were taken at least three times weekly. Euthanasia/pain scores were determined by weight loss, behavior and appearance. We only considered a score of $\geq 5$ as clinical signs attributed to viral infection because a score of 4 was often seen in non-infected controls due to weight loss alone; weight loss was not utilized as a pain score for control mice unless observed for two consecutive weight recording days. At 21 days post infection (dpi) or a pain score of 10, euthanasia was performed and gross findings during necropsy recorded. Clinical signs, mortality, and abnormal gross findings are displayed per group. Hu-CD34+-8 was excluded from this analysis since it was unsuccessfully infected.

| | | Clinical Signs | | | | | Mortality | | Necropsy Findings | | | |
|---|---|---|---|---|---|---|---|---|---|---|---|---|
| | Group | >20% Wt Loss | Minimally subdued Activity | Unresponsive when stimulated | Ruffled/ piloerection | Skin lesions | Found Deceased | Did not survive Day 21 | Hepatic necrosis | Gallbladder hemorrhage | Splenomegaly | Lymphadenomegaly |
| **hu-PBMC** | Diluent Non-infectious Control | 0/2 | 0/2 | 0/2 | 0/2 | 0/2 | 0/2 | 0/2 | 0/2 | 0/2 | 0/2 | 1/2 |
| **hu-PBMC** | γ-irradiated VARV Non-infectious Control | 0/2 | 0/2 | 0/2 | 0/2 | 0/2 | 1/2 | 1/2 | 0/2 | 0/2 | 0/2 | 0/2 |
| **hu-PBMC** | VARV $7 \times 10^5$ | 1/4 | 1/4 | 0/4 | 0/4 | 0/4 | 0/4 | 1/4 | 0/4 | 0/4 | 0/4 | 0/4 |
| **hu-PBMC** | VARV $7 \times 10^3$ | 2/4 | 0/4 | 0/4 | 2/4 | 1/4 | 0/4 | 0/4 | 0/4 | 0/4 | 0/4 | 0/4 |
| **hu-PBMC** | Negative Non-infectious Control | 0/2 | 0/2 | 0/2 | 0/2 | 0/2 | 0/2 | 0/2 | 0/2 | 0/2 | 0/2 | 0/2 |
| **hu-CD34** | γ-irradiated VARV Non-infectious Control | 0/2 | 0/2 | 0/2 | 0/2 | 0/2 | 0/2 | 0/2 | 0/2 | 0/2 | 1/2 | 0/2 |
| **hu-CD34** | VARV $7 \times 10^5$ | 4/4 | 0/4 | 2/4 | 4/4 | 0/4 | 2/4 | 4/4 | 3/4 | 0/4 | 0/4 | 0/4 |
| **hu-CD34** | VARV $7 \times 10^3$ | 3/3 | 1/3 | 0/3 | 1/3 | 1/3 | 1/3 | 3/3 | 1/3 | 2/3 | 2/3 | 1/3 |
| **hu-CD34** | Negative Non-infectious Control | 0/2 | 0/2 | 0/2 | 0/2 | 0/2 | 0/2 | 0/2 | 0/2 | 0/2 | 0/2 | 0/2 |
| **hu-BLT** | γ-irradiated VARV Non-infectious Control | 0/2 | 0/2 | 0/2 | 0/2 | 0/2 | 0/2 | 0/2 | 0/2 | 0/2 | 0/2 | 0/2 |
| **hu-BLT** | VARV $7 \times 10^5$ | 3/4 | 1/4 | 2/4 | 3/4 | 2/4 | 1/4 | 4/4 | 3/4 | 0/4 | 1/4 | 0/4 |
| **hu-BLT** | VARV $7 \times 10^3$ | 3/4 | 1/4 | 0/4 | 3/4 | 1/4 | 4/4 | 4/4 | 2/4 | 0/4 | 0/4 | 0/4 |
| **hu-BLT** | Negative Non-infectious Control | 0/2 | 0/2 | 0/2 | 0/2 | 0/2 | 0/2 | 0/2 | 0/2 | 0/2 | 0/2 | 0/2 |

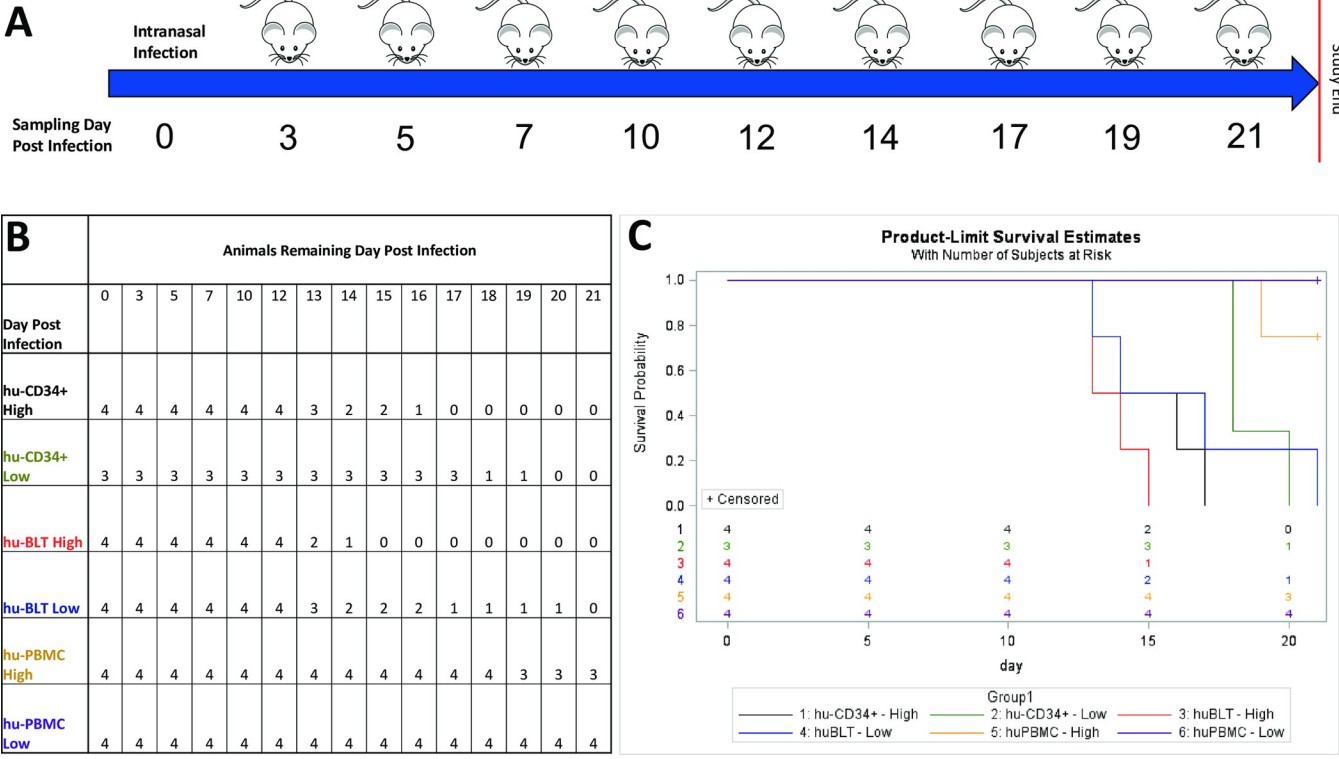

**Fig 1. Study design, survival table and survivorship curves for mice and dosage groups.** A schematic overview of the study design (A). The number of the three types of humanized mice per group based on days post infection and organized by dose group and mouse strain (B). If a sample day is not shown, there was no change in animal number per group. If a mouse was found deceased in-between PM and AM checks, the animal number was decreased the day the animal was found to have succumbed. Survival of three types of humanized mice (hu-BLT, hu-CD34+, and hu-PBMC) were assessed for low dose and high dose independently (C). *Hu-PBMC mice were significantly more likely to survive variola infection compared to hu-BLT and hu-CD34+ mice in both dose groups (high dose p = 0.004, low dose p = 0.008). There were no significant differences between hu-BLT and hu-CD34+ mice, when controlling for dose.

which may have been related to anesthesia complications; this was the only control animal to die during the study. All necropsy samples from this animal were negative for VARV DNA.

## Pathology and immunohistochemistry

Cutaneous lesions (vesicles or ulcers) were observed on the hocks of 5 mice. At necropsy, the most common gross pathologic observation was multifocal and coalescing regions of hepatic necrosis in hu-CD34+ (n = 4/7 mice) and hu-BLT (n = 5/8 mice). Sporadic splenomegaly and gallbladder hemorrhage were also seen in these strains. Hu-PBMC mice had no remarkable gross lesions (Table 1).

The following tissues were examined by histopathology and immunohistochemistry, when available: brain, lung, liver, kidney, spleen, adrenal gland, lymph nodes, bone marrow, female reproductive tract, gastrointestinal tract, oronasal tissues, and skin from hock lesions when present (the observed lesions were collected for tissue culture propagation, however the surrounding area was examined by histopathology and immunohistochemistry). Findings are summarized in Table 2 and shown in Figs 2 and 3. Hock skin lesions were identified microscopically for 1 hu-PBMC and 1 hu-CD34+ mouse, with histopathological findings including epidermal hyperplasia and hyperkeratosis with epithelial staining by VARV immunohistochemistry at the chronic ulcer margins. Periarticular tissues (synovium, tendon, periosteum) also had mild inflammation and VARV immunostaining (Fig 2A and 2B).

**Table 2. Microscopic lesions and VARV immunolocalization in tissues from humanized (PBMC, CD34+, BLT) mice and non-humanized background strain (NSG) mice inoculated intranasally with VARV.** Fractions indicate number of animals with finding divided by number of animals with tissue type collected.

| Histopathologic findings | Mouse strain, VARV dose | | | | | | | | VARV immunolocalization* |
|---|---|---|---|---|---|---|---|---|---|
| | PBMC | | CD34+ | | BLT | | NSG | | |
| | Low n = 4 | High n = 4 | Low n = 3 | High n = 4 | Low n = 4 | High n = 4 | Low n = 5 | High n = 5 | |
| Skin, hock: chronic ulceration | + (1/1) | 0 | + (1/1) | 0 | - (0/1) | - (0/2) | N/A | N/A | Hyperplastic epidermis at ulcer margin, periarticular mesenchymal cells |
| Liver: hepatocellular necrosis; rare intracytoplasmic inclusions | + (1/4) | + (1/4) | +++ (3/3) | +++ (4/4) | +++ (4/4) | +++ (4/4) | - (0/5) | - (0/5) | Hepatocytes, Kupffer cells, endothelial cells |
| Adrenal gland: multifocal cortical > medullary necrosis | + (1/1) | + (2/3) | +++ (3/3) | + (2/2) | +++ (3/3) | ++ (2/2) | 0 | 0 | Necrotic and intact cortical epithelium and medullary chromaffin cells |
| Spleen: necrosis, hemorrhage, lymphoid depletion | ++ (1/1) | - (0/1) | +++ (2/2) | +++ (3/3) | +++ (3/3) | +++ (2/2) | - (0/4) | - (0/4) | Mesenchymal cells, reticuloendothelial cells |
| Lymph nodes: necrosis, lymphoid depletion | +++ (1/4) | +++(1/1) | ++ (3/3) | + (1/1) | +++ (1/2) | +++ (2/3) | - (0/5) | - (0/5) | Reticuloendothelial cells |
| Bone marrow: necrosis, hemorrhage | + (2/4) | - (0/4) | +++ (3/3) | +++ (4/4) | +++ (4/4) | +++ (4/4) | - (0/5) | - (0/5) | Necrotic hematopoietic cells, endosteum, periosteum |
| Ovary: multifocal necrosis | ++ (2/2) | 0 | - (0/1) | - (0/2) | - (0/4) | - (0/1) | - (0/5) | - (0/4) | Stromal and follicular cells |
| Uterus: multifocal to mural necrosis | ++ (4/4) | +++ (1/4) | ++ (2/3) | ++ (1/4) | + (2/4) | + (1/3) | - (0/5) | - (0/5) | Stromal and smooth muscle cells |
| Lung: multifocal bronchial epithelial necrosis; peribronchiolar and perivascular edema; mild interstitial pneumonitis | ++ (4/4) | ++ (3/4) | + (1/3) | + (2/4) | + (3/4) | + (1/4) | - (0/5) | - (0/4) | Bronchial epithelium, interstitial cells, inflammatory cells |
| Nasal cavity: multifocal submucosal necrosis and edema, serous gland atrophy, minimal inflammation | + (3/4) | ++ (2/4) | - (0/3) | - (0/4) | - (0/4) | - (0/4) | + (1/5) | + (4/5) | Submucosal stroma, respiratory and serous glandular epithelial cells |
| Tooth: pulp necrosis | + (1/3) | + (1/4) | ++ (2/3) | ++ (1/4) | ++ (1/4) | + (2/4) | - (0/5) | - (0/5) | Pulp, periodontal ligament |
| Disseminated bacteremia | - (0/4) | - (0/4) | +++ (3/3) | +++ (4/4) | +++ (4/4) | +++ (4/4) | - (0/5) | - (0/5) | N/A |

Histopathologic scoring:—negative/not present; + mild/focal; ++ moderate/multifocal; +++ severe/extensive; 0 tissue not collected; N/A not applicable

*Immunostaining localized to foci of necrosis when present; however, immunostaining was also seen in tissues without morphologic alterations. Reported immunolocalization includes both necrotic and morphologically intact foci.

Histopathologic findings in extracutaneous tissues were generally similar between low and high dose groups for each mouse strain (Table 2 and Figs 2 and 3). Hu-CD34+ and hu-BLT mice had tissue necrosis with minimal inflammation in the liver, adrenal gland, lymphoid tissues, and reproductive tract. Abundant VARV antigen was detected within necrotic foci, but also sometimes within morphologically unaffected tissue, by immunohistochemistry. Livers showed confluent and lobular hepatocellular necrosis, with immunostaining localized prominently within necrotic and intact hepatocytes, as well as scattered Kupffer and endothelial cells. Rarely, eosinophilic globular cytoplasmic material reminiscent of viral inclusions was present within hepatocytes (Fig 3A inset). Adrenal glands had discrete foci of necrosis most commonly in the cortex, and occasionally in the medulla. Poxviral antigen localized to foci of necrosis, as well as other foci of intact cells. Examined spleens uniformly showed diffuse necrosis with severe lymphoid depletion and red pulp expansion by fibrin and hemorrhage. Poxviral antigen localized to macrophages and mesenchymal cells around central arteries in the regions of periarteriolar lymphoid sheath depletion. Lymph nodes from various sites similarly showed lymphoid necrosis and depletion, with poxviral immunostaining in reticuloendothelial cells. Bone marrow showed extensive necrosis and hemorrhage with immunostaining in the hematopoietic compartment, and

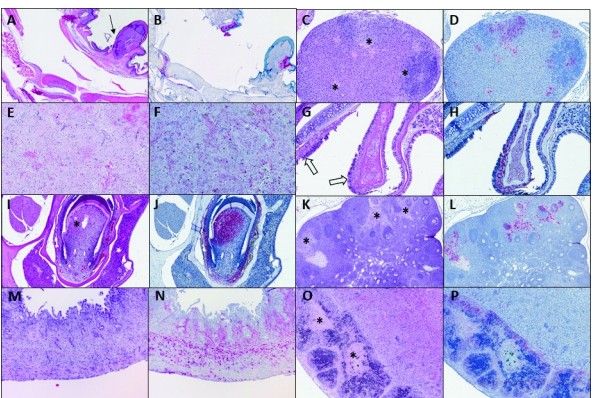

**Fig 2. Representative histopathology and immunohistochemistry of VARV infection in humanized mice. A, B**: Chronic skin ulceration over the hock, with epidermal hyperplasia (arrow, A) and VARV immunostaining in skin, tendon, and periosteum (hu-PBMC-11; low dose, day of death 21 dpi). **C, D:** Multifocal adrenal gland necrosis (*, C) with VARV immunostaining (hu-BLT-5; high dose, day of death 13 dpi). **E, F:** Diffuse splenic necrosis with lymphoid depletion, fibrin, and hemorrhage; extensive VARV immunostaining in reticuloendothelial and mesenchymal cells (hu-CD34+-4; high dose, day of death 16 dpi). **G, H**: Nasal mucosa with submucosal edema and mild inflammation (open arrows, G); extensive VARV immunostaining (hu-PBMC-6; high dose, day of death 21 dpi). **I, J:** Tooth with dental pulp necrosis (*, I); VARV immunostaining of pulp and periodontal ligament (hu-CD34+-9; low dose, day of death 20 dpi). **K, L**: Ovary with multifocal stromal and follicular necrosis (*, K) and VARV immunostaining (hu-PBMC-10; low dose, day of death 21 dpi). **M, N:** Uterus with transmural necrosis and VARV immunostaining (PBMC-5; high dose, day of death 13 dpi). **O, P**: Focal necrosis (*, O) in renal subcapsular human fetal thymic graft, with extensive VARV immunostaining (hu-BLT-5; high dose,). A, C, E, G, I, K, M, O (hematoxylin-eosin); B, D, F, H, J, L, N, P (VARV immunohistochemistry, viral antigen labeling in red). Original magnifications: A, B, O, P (x50); C-J, M, N (x100); K, L (x200).

patchy staining of endosteum and periosteum. Lungs had a mild increase in interstitial cellularity, with scattered interstitial immunostaining. One hu-BLT mouse (hu-BLT-6) also had multifocal bronchiolar epithelial necrosis and immunostaining. Nasal tissues had scattered small foci of submucosal, and rarely respiratory epithelial or serous glandular, immunostaining, without apparent inflammation or necrosis. Teeth had multifocal to extensive immunostaining that was concentrated in the dental pulp, sometimes in association with necrosis, and the periodontal ligament. For available female reproductive tract tissues, the ovarian stroma consistently had scattered viral immunostaining, without overt morphological alterations; follicles also stained to a lesser extent. 2/7 hu-CD34+ uteri, and 3/7 hu-BLT uteri, had moderate to extensive immunostaining of smooth muscle and stromal (including perivascular) cells, variably accompanied by necrosis. Oviduct and vagina had scattered or patchy immunostaining in a pattern similar to that observed in the uterus, without overt necrosis. Gastrointestinal tissues also showed inconsistent and rare, scattered staining in the smooth muscle, serosa, and rarely submucosa. Kidneys showed very rare, scattered immunostaining within glomeruli and interstitial cells. For hu-BLT mice with remnant subcapsular grafts, grafted human tissues showed extensive immunostaining. Brain had no histopathologic findings and no immunostaining, and heart was not available, from any animal of these strains. Disseminated bacteremia was seen histologically in all virally challenged animals of these two strains, with bacterial emboli consistently found in brain, lung, liver, spleen, and kidney. Immunohistochemical testing of a subset of tissues from four animals (2 hu-CD34+ and 2 hu-BLT) detected involvement of multiple, mixed bacteria, including *Staphylococcus spp.*, *Enterococcus spp.*, and *Streptococcus spp.* Gram-negative bacteria were not identified. Control animals inoculated with phosphate-buffered saline (PBS) or gamma-irradiated virus had no significant histopathologic findings and no VARV immunostaining (S1 Fig).

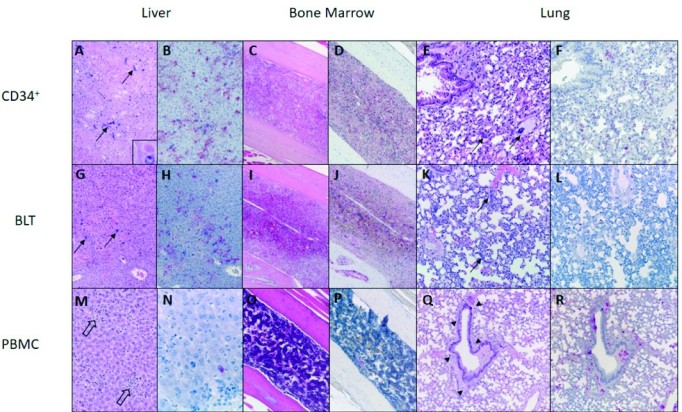

**Fig 3. Representative differences in pathologic findings among three strains of humanized mice with VARV infection.** Hu-CD34+ and hu-BLT mice had similar findings in liver, bone marrow, and lung, which contrasted those seen in hu-PBMC mice. Hu-CD34+ and hu-BLT livers (A, G) had confluent and lobular hepatocyte necrosis with very rare eosinophilic globular inclusions (A, inset), and VARV immunostaining of hepatocytes, Kupffer cells, and occasional endothelial cells (B, H). Hu-CD34+ and hu-BLT bone marrow specimens similarly showed extensive necrosis and hemorrhage associated with VARV immunostaining (C, D, I, J). Hu-PBMC liver (M) and bone marrow (O) showed minimal inflammation (open arrows, M) and no necrosis, and only very rare VARV immunostaining in these tissues (N, P). Conversely, hu-CD34+ and hu-BLT lung tissues showed minimal inflammation (E, K) and VARV immunostaining (F, L), while hu-PBMC lungs (Q) had more prominent peribronchiolar and perivascular inflammation (arrowheads, Q) with VARV immunostaining (R). Hu-CD34+ and hu-BLT mice had disseminated intravascular bacteria (arrows in A, E, G, K), which were not present in hu-PBMC mice. A, B, E, F (hu-CD34+-4, high dose, day of death 16 dpi); C, D (hu-CD34+-6, high dose, day of death 17 dpi); G, H (hu-BLT-5, high dose, day of death 13 dpi); I, J (hu-BLT-9, low dose, day of death 17 dpi); K,L (hu-BLT-4, high dose, day of death 13 dpi); M, N, Q, R (hu-PBMC-11, low dose, day of death 21 dpi); O, P (hu-PBMC-5, high dose, day of death 19 dpi). A, C, E, G, I, K, M, O, Q (hematoxylin-eosin); B, D, F, H, J, L, N, P, R (VARV immunohistochemistry, viral antigen labeling in red). Original magnifications: A, B, C, D, G, H, I, J, O, P (x100); E, F, K, L, Q, R (x200); M, N (X400).

Hu-PBMC mice had overall similar findings, but with less liver, bone marrow, and tooth involvement, increased nasal and lung involvement, slightly more prominent inflammation overall, and absence of bacteremia (Table 2 and Figs 2 and 3). Lungs from hu-PBMC mice had more consistent and numerous foci of bronchiolar epithelial necrosis accompanied by perivascular and peribronchiolar edema and mild inflammation, and mild interstitial pneumonitis. Immunostaining localized to bronchiolar epithelial cells, peribronchiolar and perivascular stromal and inflammatory cells, and scattered interstitial cells. Nasal tissues had patchy edema, necrosis, and mild inflammation, with necrosis and atrophy of serous glands, with corresponding multifocal to widespread immunostaining of epithelium and submucosal stroma and glands. These changes were more prominent in the high dose hu-PBMC group. One animal in the low dose group (hu-PBMC9) had a fibrinocellular and hemorrhagic exudate, which showed granular intra- and extracellular VARV immunostaining, in the middle nasal meatus. All uninfected, and n = 2/8 infected hu-PBMC animals had systemic, atypical lymphoid proliferation suggestive of the development of graft vs host (GVH) disease. Control animals had no VARV immunostaining (S1 Fig).

A subset of tissues were examined by transmission electron microscopy (Fig 4). Ultrastructural evaluation revealed almost exclusively immature VARV particles in the hepatocytes of the native mouse liver tissue of infected hu-CD34+ (n = 2), hu-BLT (n = 2) and hu-PBMC (n = 1) mice (Fig 4A.); however, both mature and immature particles were located in the sinusoidal endothelial cells and free in the sinusoids of the livers (Fig 4B). In the examined hu-PBMC murine ovary, uterus, and adventitial cells surrounding the bile duct, and in human fetal thymic allograft from a hu-BLT mouse (Fig 4C), both mature and immature particles were present.

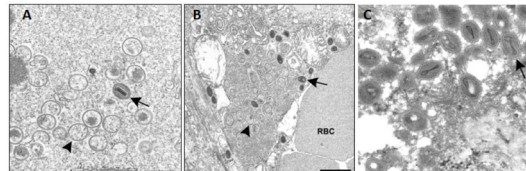

**Fig 4. Electron microscopic images of VARV in a humanized mouse.** Subset of select tissues from VARV challenged humanized-mice (hu-BLT, hu-CD34+ and hu-PBMC) were examined by transmission electron microscopy. (A) Hepatocyte with multiple spherical, diffuse immature particles (arrowhead) and a single condensed mature particle (arrow). Bar, 500 nm. (B) Sinusoidal endothelial cell in the liver with immature (arrowhead) and mature (arrow) VARV particles. RBC, red blood cell. Bar, 1 um. (C) Human fetal thymic allograft containing mature virions with a characteristic dumbbell-shaped nucleoid (arrow). (BLT-8, low dose, day of death 13 dpi). Bar, 100 nm.

## Molecular findings

Of the 222 oral swabs that were collected throughout the study, only 4 were positive for viral DNA. Two oral swabs contained viable virus and were from the lower dose groups: hu-BLT-8 on 12 dpi (46.2 pfu/ml) and hu-PBMC-11 on 19 dpi (594 pfu/ml). All hock lesion swabs were negative for viral DNA, but all five hock lesion tissues contained viral DNA and viable virus (Fig 5). Remarkably high loads of viable virus were present in multiple tissues, including heart, kidney, liver, lung, ovaries and spleen, and in some instances reached as high as $1.66 \times 10^{11}$ pfu/gram of tissue (Fig 5). The hu-BLT mice had the highest viral loads (both dose groups) followed by slightly lower levels in hu-CD34+ mice. Tissue viral load comparisons between hu-BLT and hu-CD34+ were not significantly different when controlling for dose. Despite little mortality in the hu-PBMC mice, high viral loads were detected in most of the tissues tested. Hu-PBMC-5, a mouse in the hu-PBMC high dose group, had the highest viral loads compared to other mice in that group and was the only hu-PBMC euthanized (19 dpi) due to clinical signs. Minimal whole blood euthanasia samples were available for evaluation of viremia. Viral DNA was detected in 9/10 infected animals (3/4 high dose hu-PBMC, 4/4 low dose hu-PBMC, 2/2 high dose hu-BLT) ranging from $1.67 \times 10^2$ to $2.05 \times 10^5$ fg/μl with hu-BLT mice having the highest quantities. Post DNA evaluation, three blood samples had sufficient volume remaining for viral titration, all from the hu-PBMC low dose group, and 2/3 contained viable virus ($2.0 \times 10^2$ and $1.39 \times 10^3$ pfu/ml). To look for evidence of antibodies post infection, serum (when available) was tested for the presence of human IgM and IgG by ELISAs. Serum available for testing included all hu-PBMC mice (excluding oneγ-irradiated control animal), all hu-CD34+ control mice, two from the high and two from the low dose group, and for hu-BLT mice only serum from the γ-irradiated VARV control group, one uninfected animal and four from the high dose group. None of the tested serum samples had detectable levels of human IgM or IgG in ELISAs designed for human serum.

## Non-humanized NSG mice are not susceptible to systemic VARV disease

Part two of the study determined whether the immunosuppressed NSG background mouse was susceptible to systemic VARV infection. Clinical signs and weight loss were minimal/absent in the NSG mice (both doses) and all NSG animals survived until study end (21 dpi). No abnormal findings were seen during necropsy. On pathologic evaluation, 1/5 in the low dose group, and 4/5 in the high dose group had very mild inflammatory changes associated with VARV immunostaining in the nasal submucosa (S2 Fig). No significant histopathologic changes, and no VARV immunostaining, were seen in extranasal tissues (lung, liver, spleen, kidney, lymphoid tissues, and female reproductive tract) (S2 Fig). The oral and skin swabs

### hu-BLT 7x10³

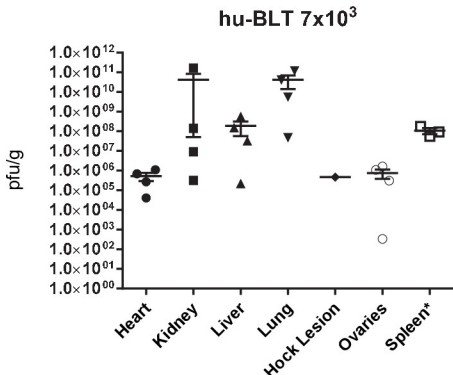

### hu-BLT 7x10⁵

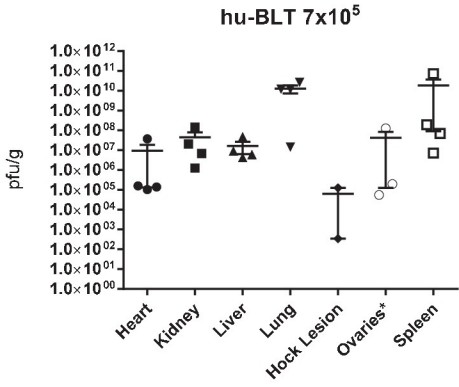

### hu-CD34⁺ 7x10³

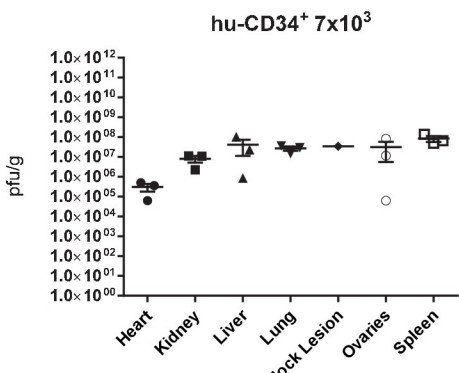

### hu-CD34⁺ 7x10⁵

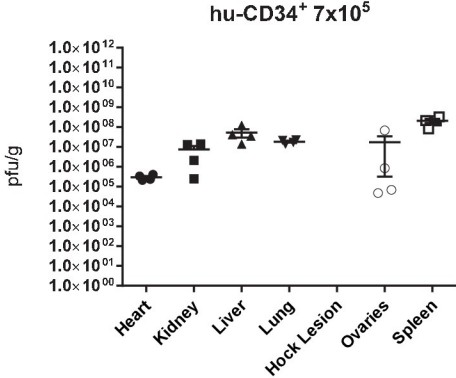

### hu-PBMC 7x10³

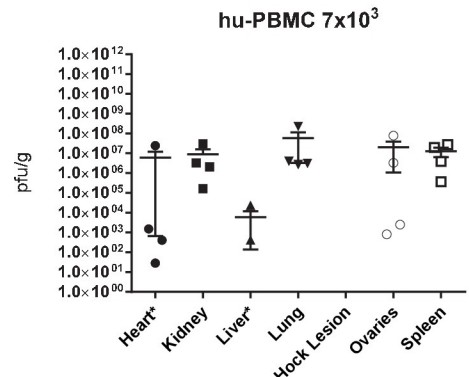

### hu-PBMC 7x10⁵

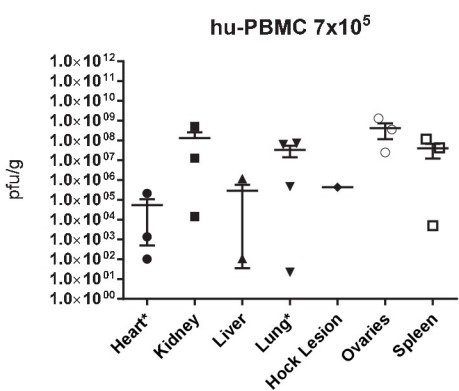

**Fig 5. High viable virus loads were detected in all three types of humanized mice.** Humanized-mice (hu-BLT (A), hu-CD34+ (B) and hu-PBMC (C)) were inoculated via the intranasal route with $7 \times 10^3$ or $7 \times 10^5$ plaque forming units (pfu) of VARV (VARV_JAP51_hrpr (primary clade I)) (n = 4 per group). Animals that succumbed to VARV infection, or were euthanized at 21 dpi, had tissues collected and processed for viral titration (plaque assay). The mean with SEM is shown. An * indicates one or more of that sample had cell culture monolayer destroyed or plaques were present but below the limit of detection for this assay.

were negative for viral DNA. Several necropsy tissues, mainly the lung, liver and nasal cavity, from several NSG mice were positive for viral DNA (S3 Fig). However, the only tissues that contained viable virus were two nasal cavities, one from the low and one from the high dose group, which was the site of inoculation (S3C Fig). All NSG negative control mice were negative for viral DNA and had no significant pathologic findings and no VARV immunostaining (S1 Fig).

We utilized hu-PBMC mice as positive controls during this part of the study; similar to part one of the study, hu-PBMC mice began displaying clinical signs late in the study (~day 19) and one mouse had to be euthanized on day 19. Molecular and pathology results were overall similar to part one of the study, with the virus spreading systemically and high loads of viable virus found throughout most tissues tested (S3D Fig).

## Discussion

Due to the discontinuation of smallpox vaccination after disease eradication, the human population is increasingly susceptible to smallpox. Its malicious release could have devastating consequences, making imperative the continued development of preventive and therapeutic countermeasures. Animal models are an invaluable tools for the development of MCMs. For smallpox MCM testing, an ideal animal model would mimic natural human smallpox disease using the authentic agent (VARV), use a realistic infectious dose and aerosol droplet route of infection, have an 8–14 day incubation period, identifiable prodrome, detectable immune response, high mortality and systemic rash 1–4 days post prodrome [7,8]. Various models have successfully fulfilled parts, but not all, of this ideal. Historically, most adult animals have been insusceptible to VARV challenge, even when susceptible to other orthopoxviruses. While CAST/EiJ mice and prairie dogs (*Cynomys ludovicianus*) are susceptible to an IN *Monkeypox virus* challenge [9–12], neither developed a systemic infection following IN VARV challenge [13,14]. Similar findings were observed in ICR and SCID mice following IN VARV challenge [15]. NHPs are susceptible to VARV infection, with development of systemic rash illness and mortality; however, this model has several disadvantages including: use of infectious doses much higher ($1 \times 10^8$ to $1 \times 10^9$ pfu) than the suspected dose required for human infection, and requires intravenous inoculation, an unnatural route of infection which bypasses the initial local replication and viremia, eliminating the incubation and prodromal periods [16]. In this model, only infectious doses at $1 \times 10^9$ pfu resulted in high mortality which manifested as hemorrhagic smallpox with animals succumbing to the disease as early as 4 dpi [16,17] limiting the window for testing post-exposure MCMs. Due to these difficulties, animal models using surrogate orthopoxviruses have been utilized to test MCMs, but none against the authentic agent VARV.

Here we present the first small animal models of human smallpox, utilizing humanized mice that are highly susceptible to VARV infection. We have shown that these novel humanized mouse models will be useful for studying VARV infection and testing efficacy of MCMs against the virus. While all three humanized mouse strains were susceptible to VARV and supported a productive infection, the more advanced hu-CD34+ and hu-BLT mice are the best candidates for further model characterization. The engraftment of human CD34+ cells in the hu-CD34+ mouse results in the development of a self-renewing naïve human immune system located in the mouse bone marrow as well as circulating in the tissues; it is comprised of multiple human

cells such as T, B, natural killer and antigen presenting cells [18] and the lymphocytes are H2 class restricted [18,19]. Further advanced, the hu-BLT is described as a complete immune system with the engraftment of human CD34+ cells and additionally, engraftment of human fetal and liver tissue allowing for the development of not only a variety of human cells but a naïve human immune system that is HLA restricted [18,20] in the mouse. For both strains, high and low doses of VARV administered IN produced systemic disease and high mortality, with pathologic features resembling aspects of the severe, highly lethal hemorrhagic form of human smallpox [8] and also corroborated what has been shown in high dose intravenous inoculation of VARV in macaques [16,17]. Pertinent features include hepatosplenic, lymphoid, and hematopoietic necrosis, with widespread VARV antigen and nucleic acid detection in these and other tissues, and the uniform presence of bacteremia with a variety of gram-positive cocci at the time of death. Systemic bacterial infections due to gram-positive cocci are a common feature of fatal human smallpox [8,21,22], and were also uniformly seen in intravenously inoculated macaques that developed hemorrhagic disease [17]. The role of bacterial infections as potentiators of VARV infection and/or secondary infections being the immediate cause of death in human smallpox has been debated [8,21,23–26], and these mice models may be valuable in investigating this important aspect of severe human smallpox. Although the model developed pathologic features resembling aspects of the hemorrhagic form of human smallpox, the approximate 13-day incubation period in these mice better approximates that of ordinary human smallpox, and allows a broader window for testing potential new MCMs than some other animal models which have a more rapid mortality onset [7,16].

Cutaneous lesions (vesicles and ulcers) were occasionally, but not consistently, identified in these mice, suggesting that their disease progression may most closely resemble that of early hemorrhagic smallpox, in which severe disease develops before cutaneous vesicular lesions. It could be that the animals did not live long enough for a systemic rash to form, or subtle skin lesions were obstructed by the animal's fur. The presence of infectious virions in the skin was not evaluated in this study (with exception of the collected hock lesions) and should be examined in future studies. Moreover, hemorrhage per se was not a feature of VARV infection in these mice and therefore, disease in this model may not fit neatly into one of the well-defined clinical types of human smallpox [8,16,17]. As with other animal models, the time from onset of clinical signs to death was short in these mice. A defined prodrome was not apparent, and antemortem noninvasive samples (oral swabs, skin swabs) were not useful for confirmation of infection. The absence of mucocutaneous exudates and lack of deroofing prior to swab collection may account for the futility of these samples. Further studies with lower doses of VARV are warranted to attempt to both recreate ordinary smallpox and to increase the prodromal period and potentially identify distinct biomarkers of infection that may be useful when evaluating new antivirals in a post exposure setting. Additional antemortem blood sampling should be considered in future studies as a method for confirmation of infection prior to development of clinical signs and to evaluate hematologic alterations in disease as well as determine if any biomarkers predict outcome. While human IgG and IgM were not detected in the limited serum samples available in this study, future studies should include other detection methods for evaluation of human immune response which would strengthen the model. Subsequent studies to evaluate the detection of human immune response (in particular the hu-BLT model since IgG has been reported [27–29]), as well as studies to understand the viral trafficking of the virus in the host will provide greater understanding of the utility of the model to evaluate MCMs as well as potentially provide valuable samples to understand the human immune response in these animals.

Of the assessed strains of humanized mice, the hu-PBMC mice were deemed the least promising for studying systemic VARV infection. This was based on their apparent delayed

disease course evidenced by little mortality despite high viral loads in tissue, and the development of systemic lymphoproliferative lesions, attributed to GVH [19]. GVH develops in all three of these humanized mice strains and has been well described elsewhere [18–20], but onset is earlier (within 4 weeks) in hu-PBMC mice, compared to 20 weeks and 12 months in hu-BLT and hu-CD34+, respectively. However, the hu-PBMC model may be useful for investigating specific aspects of VARV disease, such as respiratory or reproductive tract effects based on histopathology results. Additionally, comparing the hu-PBMC mice to the non-humanized NSG background mice is useful in hypothesizing why the humanized mice are susceptible to VARV infection as it is the least humanized mouse strain used in this study. The non-humanized, immunosuppressed NSG background mice lack an adaptive immune response and have less functional innate responses compared to other immunodeficient mice [19]. We found that after VARV challenge, the NSG mouse supported virus replication only at the site of inoculation (viable virus found in nasal tissue in 2/10 mice at 21 dpi) but was unable to disseminate to additional tissues. The NSG mice are engrafted with human PBMC cells (to create the hu-PBMC mouse strain) resulting in the development of primarily mature human donor HLA-restricted T cells (poor engraftment has been reported of monocytes and B cells [18]). Human effector and memory T cells can be found in secondary lymph organs and circulation [18,20] suggesting VARV infects human T cells and that infection is sufficient to allow for a productive VARV infection in these mice. We can further hypothesize that VARV uses the T cells to spread systematically in the host and replicates in human T cells and host mouse tissue as well. This is supported by detectable viremia and comparable viable virus loads in lymphoid and non-lymphoid organs. Future time course studies should be conducted to determine how the virus traffics in the mouse and what human cell populations are present focusing on early time points. Taken together, it appears that unless VARV is given at extremely high infectious doses intravenously as seen in the VARV NHP model, some component of the human immune system is needed in order to spread systemically.

Of note, compared to traditional laboratory mice, the humanized mice are more variable due to the humanization process, which is a potential limitation in regard to the reproducibility as an animal model. Although not an outbred model the humanization levels differ for each mouse [30]. However it's important to note that quality control (QC) is performed (by the company) on the human marker levels using flow cytometry before animals are approved for use in studies. Due to the more advanced humanization methodology used to create the hu-BLT mice, which includes engraftment of human tissue within the mouse, it is reasonable to assume that the variation within this mouse strain is most pronounced. This was observed when comparing the viral titers between the hu-BLT mice, which did vary. Hu-CD34+ mice were more similar in regard to viral load, which might be explained by the more simplified humanization process which does not entail human tissue engraftment. However, tissue viral load comparisons between hu-BLT and hu-CD34+ were not significantly different when controlling for dose. The relatively consistent titers seen in the hu-CD34+ may be a benefit of this mouse strain for future studies. Regardless of trends in titer variation, for both hu-BLT and hu-CD34+ mice, the end result of mortality was the same for all mice. The high VARV challenge dose for both mouse strains resulted in similar days until death while more variation was seen in the time until death for the hu-BLT group challenged with a low dose. The hu-CD34 + mice challenged with a low dose had a consistent time until death, like that seen with the high challenge group. The observed 100% mortality for both mouse strains make them suitable for the purpose of therapeutic testing regardless of any variation in viral titers. Hu-PBMC mice viral titers are harder to compare due to delayed viral disease progression and viral dissemination; however, we are not considering these for future studies due to GVH as mentioned above.

We hypothesize the development, prevalence and location of human immune cells within these three mouse strains [18,20] are contributing to the differences we observed in the time course of disease, with the least humanized (hu-PBMC) mice having a delay in clinical signs and mortality. *In vitro* studies with ECTV using mouse primary cell lines such as macrophages and fibroblasts have pointed towards an important role of the mitochondria during viral replication and morphogenesis [31–33]. Performing similar VARV *in vitro* studies with different human immune primary cell lines, or human CD34+ material that is used to engraft the mice, to compare both viral progression and gene expression within the cells would be informative. Using the findings from our *in vivo* studies combined with these proposed additional in vitro studies could help in determining the differences observed between the three humanized mice strains and importantly which human cell(s) are needed for VARV infection when the inoculum is not given as a high dose via a systemic (e.g. intravenous) route.

Numerous studies have been performed looking at the VARV genome in order to help in understanding why this virus is solely a human pathogen [34–37]. When analyzing the VARV genome and other closely related poxviruses, authors detected a "hotspot" of genome variation within the VARV ortholog of the vaccinia virus O1L gene, at the level of single nucleotide polymorphisms [38]. The O1L gene has been shown to be necessary for efficient replication of *Vaccinia virus* in human cells [39]. This gene is non-functional in the two most closely related viruses to VARV (*Camelpox virus* and *Taterapox virus*), which rarely infect humans, and typically cause self-limiting infections when they do [38,40,41]. Authors note that the VARV O1L ortholog has been retained in all VARV isolates, therefore it is possible that the gene has undergone subsequent selection for optimal function in human cells and may be critical for dissemination and establishment of VARV within human hosts. When investigating the evolution of the VARV genome, investigators recently hypothesized genetic losses that have occurred during evolution of the virus [42]. Comparison of the genetic differences between modern VARV isolates compared to ancient isolates could identify potential genetic elements involved in the evolution of smallpox to becoming a solely human pathogen. In addition to these types of genetic studies, this newly characterized humanized mouse model could allow identification of human components necessary for VARV systemic spread. Further studies, as suggested above, should be done to explore which human characteristic(s) contribute to a productive infection in these humanized mice.

In conclusion, our results indicate that hu-CD34+ and hu-BLT mice will be valuable as models of human smallpox for continued development of pre- and post-exposure treatments. There is currently only one antiviral compound licensed by the U.S. FDA (TPOXX, SIGA) for treatment of smallpox infection. Previous *in vivo* studies have found that multi-drug treatment (ST-246 and CMX-001) has higher levels of protection from mortality than single drug therapy [43,44] and resistance has been reported with single-drug treatment [45]. The WHO Advisory Committee on Variola Virus Research (ACVVR) has recommended the licensure of at least two therapeutics for treatment of smallpox prior to destruction of all viral stocks. While surrogate models of smallpox infection have utility, they lack *in vivo* testing against the authentic agent of smallpox, VARV. Additionally, surrogate animal models do not allow investigation of the role of the human immunologic response in VARV infection. Our results showed that non-humanized NSG mice were not susceptible to systemic disease, indicating that a human component is required for the virus to spread from the inoculation site and cause severe smallpox disease development. Future studies that further characterize and develop these models may both strengthen the correlation of animal model and human disease (i.e. aid in identifying why VARV is solely a human pathogen), and also provide the novel opportunity to investigate the role of human-specific immunologic responses in VARV infection and smallpox disease.

## Methods

### Ethics statement

Animal care and use procedures were approved and followed according to the U.S. CDC Institutional Animal Care and Use Committee (IACUC) under protocol number 2671GALMOUC.

### Mice

Female Hu-PBMC, hu-CD34+ and hu-BLT (NSG background) and NSG mice, were purchased from The Jackson Laboratory (JAX) (additional information can be found: hu-BLT [46] and on the JAX website for hu-PBMC and hu-CD34+: https://www.jax.org/). Hu-CD34 + and hu-BLT were received at 15 weeks old (12 weeks post engraftment) while hu-PBMC mice were 8 weeks old (3 days post engraftment) during part one of the study. For part two, NSG mice were received at 7 weeks old and hu-PBMC mice were 7 weeks old (~1 week post engraftment). Precautions were taken including the use of sterile drinking water and sterilization by autoclaving of bedding, enrichment and food used during the study. Mice were grouped housed (n = 3–5) upon being received and acclimated for at least 3 days prior to beginning the studies. After acclimation but prior to inoculation, serum was collected via the submandibular vein (U.S. CDC IACUC policy 026).

### Challenge virus

Work with live VARV is conducted in the Biosafety Level 4 laboratory (BSL-4), approved by the WHO ACVVR, is done in accordance with all applicable U.S. Federal Select Agent regulations (42 CFR part 73) and all inactivation procedures used had been reviewed and approved by the U.S. CDC Laboratory Safety Review Board. These studies are reviewed biannually as part of the United States Government Policy for Oversight of Life Sciences Dual Use Research of Concern.

A semi-purified preparation of VARV_JAP51_hrpr (primary clade I) was selected as the challenge virus, as it has been used in historic NHP studies [16,17,47] as well as other mouse studies [13]. This virus was isolated from a U.S. solider in Japan in 1951 and was part of the U. S. Army repository before being transferred to the U.S. CDC VARV collection [16]. The isolate underwent 5 passages on chorioallantoic membrane before being passaged twice in African green monkey kidney cells (BSC-40) and undergoing purification as previously described [48–53]. Inoculum was diluted in PBS. For γ-irradiation inactivation, the virus was treated with 5 x $10^6$ rads. All challenge doses were confirmed via standard cell culture techniques (back-titration) immediately following inoculation.

### Animal inoculation

All mice were inoculated via the IN route to mimic the natural route of human smallpox infection. Viral inoculation was done while animals were maintained under 3–5% inhalation isoflurane anesthesia. Inoculum was diluted in PBS and 0.05% bovine serum albumin. Animal group sizes are defined as "n". For part one of the study, Hu-PBMC, hu-CD34+ and hu-BLT mice were inoculated with $7x10^3$ or $7x10^5$ plaque forming unit (pfu) (n = 4 per group). To serve as controls, two hu-PBMC mice were inoculated with diluent and two of each mouse strain were inoculated with γ-irradiated VARV_JAP51_hrpr equivalent to $7x10^5$ pfu. For part two of the study, NSG mice were inoculated with $4x10^6$ or $5x10^4$ pfu (n = 5 per group). For negative controls, NSG mice were uninfected or mock-infected with diluent (n = 3 per group). Three hu-PBMC mice were also included in the part two of the study and challenged with $4x10^6$ pfu to serve as positive controls.

## Animal sampling, observations and euthanasia

Clinical signs were recorded daily, and oral swabs, physicals and weights were taken three times weekly under 3–5% inhalation isoflurane anesthesia. Euthanasia/pain scores were determined by weight loss, behavior and appearance, with clinical scoring performed twice daily once animals reached a score of $\geq$5. We only considered a score of $\geq$5 as clinical signs attributed to viral infection because a score of 4 was often seen in non-infected controls due to weight loss alone; weight loss was not utilized as a pain score for control mice unless observed for two consecutive weight recording days. At 21 dpi or a pain score of 10, euthanasia was performed via exsanguination and cervical dislocation under 5% inhalation isoflurane anesthesia. Oral swab (polyester frozen dry), tissues, whole blood and serum (Starstedt) were collected upon euthanasia. The following tissues were collected: nose, lung, liver, spleen, ovaries, heart, kidney and any other tissue with abnormal appearance. Between samples, necropsy tools were decontaminated in 5% micro-chem, scrubbed with a brush and rinsed with water. All samples were frozen at -80˚C for future analysis.

## Sample processing and DNA extraction

Swabs were processed as previously described [10]. Tissues and whole blood were thawed on ice. Aliquots of 1 mm zirconia/silica beads (Biospec) and PBS for tissue homogenizing were made by mixing 0.5 ml and 0.7 ml, respectively. Each beads/PBS tube was weighed prior to pouring contents into the tissue tube; the tissue tube was weighed and change in readings was used as an approximate tissue weight. Samples were homogenized using the Mini-BeadBeater-16 (Biospec) by grinding twice for one minute, and icing samples for one minute between runs. Sample inactivation (per internally validated/approved inactivation method) and DNA extraction were performed as previously described except samples were heated at $\geq$56˚C [10]. The remaining sample was re-frozen at -80˚C.

## Pathology and immunohistochemistry

During necropsy, a portion of each collected tissue and the remaining mouse carcass were placed into 10% neutral-buffered formalin for 7 days for virus inactivation (per internally validated/approved inactivation method) and tissue fixation and were transferred to a BSL-2 laboratory. Tissues were processed for routine paraffin histology, and sections were cut at 4 microns and stained with hematoxylin-eosin (H&E). Immunohistochemical (IHC) detection of VARV was performed as previously described [54], using a rabbit polyclonal anti-VARV antibody (U.S. CDC, Atlanta, GA). Formalin-fixed, ɣ-irradiated, paraffin-embedded monkey kidney cells infected with VARV were used as a positive control. Negative control utilized normal rabbit serum in place of the primary antibody. Tissues for electron microscopic examination were either formalin-fixed and placed in phosphate buffer and then buffered with 2.5% glutaraldehyde or were processed by the on-slide embedding technique [55]. Samples were then embedded in Epon/Araldite by standard methods [56].

## Molecular assays

Real-time PCR was performed as previously described [57]. A standard dilution series of semi-purified VARV DNA was assayed on each PCR plate in order to quantify DNA levels within samples. Two fg viral DNA/rxn was the positive cut-off value. Samples that crossed the threshold ($C_t$ value) before cycle 38 in duplicate underwent a modified version of titration with a 96 hour incubation [10]. Samples were thawed on ice prior to sonication at 40% output twice at one-minute intervals, with 10 seconds of icing between sonication. Samples were titrated on BSC-40 cells with

a 96hour incubation prior to staining with crystal violet stain. All samples were serially diluted and titrated in duplicate. The following quality control measures were used to determine if sample needed to be repeated: $\geq$25% of the CV stained monolayer was missing in the countable wells, if samples did not dilute out as anticipated when conducting serial dilutions, and if counts could not be determined in any dilution due to diluting out too far. The limit of detection for this assay was pre-determined as an average of at least $\geq$3 plaques in the lowest dilution ($10^{-1}$). Detection of anti-Orthopoxvirus IgG and IgM was performed as previously described [58].

## Statistics

Comparisons of the tissue viral load (pfu/ml) between mouse strains and dose were made using the Wilcoxon rank-sum test, as the data are not normally distributed. Differences in mortality rates for three humanized mouse strains were compared using Fisher's exact test, due to small sample size. Kaplan-Meier survivorship curves were calculated and compared using the log-rank test. A p-value $< 0.05$ was considered statistically significant. Data analysis was performed using SAS version 9.4 (SAS Institute).

## Supporting information

**S1 Fig. VARV immunohistochemistry in representative tissues from control mice of each strain inoculated with PBS only or with gamma-irradiated VARV.** No VARV immunostaining (viral antigen labeling in red) is present in liver, kidney, lung, or bone marrow from hu-PBMC mouse inoculated with PBS only (top row, hu-PBMC-1, day of death 21 dpi); hu-PBMC mouse inoculated with gamma-irradiated VARV (second row, hu-PBMC-3 day of death 13 dpi); hu-CD34+ mouse inoculated with gamma-irradiated VARV (third row, hu-CD34+-1); hu-BLT mouse inoculated with gamma-irradiated VARV (fourth row, hu-BLT-1, day of death 21 dpi); NSG mouse inoculated with gamma-irradiated VARV (fifth row, NSG-11, day of death 21 dpi); NSG mouse inoculated with PBS only (bottom row, NSG-16, day of death 21 dpi). Original magnifications: liver, kidney, lung (x40); bone marrow (x100). (TIF)

**S2 Fig. Representative histopathology and VARV immunohistochemistry in non-humanized NSG mice.** A, B: Nasal tissue with mild inflammatory changes and immunostaining of VARV (NSG-9, high dose, day of death 21 dpi). C, D: Liver without histopathologic changes or immunostaining of VARV (NSG-9, high dose). E, F: Lung without histopathologic changes or immunostaining of VARV (NSG-5, low dose). A, C, E (hematoxylin-eosin); B, D, F (VARV immunohistochemistry, viral antigen labeling in red). Original magnification: all images (x100). (TIF)

**S3 Fig. NSG background mouse was not susceptible to systemic variola virus infection.** NSG mice were inoculated with $4x10^6$ or $5x10^4$ pfu for a high and low dose group (n = 5 per group). Three hu-PBMC mice were also included and challenged with $4x10^6$ pfu to serve as positive controls. On 21 dpi, animals were euthanized. (A,B): Several necropsied tissues from NSG mice contained low levels of viral DNA mainly the lung, liver and nasal cavity in both challenge groups. (C) Only two nasal cavities, one animal from the low (NSG-1) and one from the high dose group (NSG-6) contain viable virus. An $^*$ indicates one or more of that sample had cell culture monolayer destroyed or plaques were present but below the LOD for this assay. (D) In contrast, virus spread systemically through the n = 3 hu-PBMC mice leading to clinical signs and one animal had to be euthanized on day 19 due to a clinical score of 10; high loads of viable virus were found throughout most tissues tested from the hu-PBMC mice. (TIF)

## Acknowledgments

Michael Townsend, Jillybeth Burgado, Brock Martin, Zachary Weiner, Scott Smith, Theodora Khan, Todd Smith and Felix Jackson for various supportive roles throughout the study; and Heather Hayes and Brigid Bollweg for histology and immunohistochemistry support.

The findings and conclusions in this report are those of the authors and do not necessarily represent the official position of the Centers for Disease Control and Prevention.

## Author Contributions

**Conceptualization:** Christina L. Hutson, Ashley V. Kondas, Nadia Gallardo-Romero, Darin Carroll, Victoria A. Olson.

**Data curation:** Christina L. Hutson, Ashley V. Kondas, Jana M. Ritter, Clint N. Morgan, Christine M. Hughes.

**Formal analysis:** Christina L. Hutson, Ashley V. Kondas, Jana M. Ritter, Clint N. Morgan, Christine M. Hughes.

**Funding acquisition:** Darin Carroll, Victoria A. Olson.

**Investigation:** Christina L. Hutson, Ashley V. Kondas, Jana M. Ritter, Zachary Reed, Sharon Dietz Ostergaard, Clint N. Morgan, Nadia Gallardo-Romero, Cassandra Tansey, Matthew R. Mauldin, Johanna S. Salzer, Cynthia S. Goldsmith, Victoria A. Olson.

**Methodology:** Christina L. Hutson, Ashley V. Kondas, Jana M. Ritter, Zachary Reed, Nadia Gallardo-Romero, Cassandra Tansey, Christine M. Hughes, Cynthia S. Goldsmith, Victoria A. Olson.

**Supervision:** Christina L. Hutson, Darin Carroll, Victoria A. Olson.

**Validation:** Christina L. Hutson, Ashley V. Kondas, Jana M. Ritter, Clint N. Morgan, Christine M. Hughes, Cynthia S. Goldsmith.

**Visualization:** Christina L. Hutson, Ashley V. Kondas, Jana M. Ritter, Christine M. Hughes, Cynthia S. Goldsmith.

**Writing – original draft:** Christina L. Hutson, Ashley V. Kondas, Jana M. Ritter, Clint N. Morgan, Christine M. Hughes.

**Writing – review & editing:** Christina L. Hutson, Ashley V. Kondas, Jana M. Ritter, Zachary Reed, Sharon Dietz Ostergaard, Clint N. Morgan, Nadia Gallardo-Romero, Cassandra Tansey, Matthew R. Mauldin, Johanna S. Salzer, Christine M. Hughes, Cynthia S. Goldsmith, Darin Carroll, Victoria A. Olson.

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
