## [Decision Letter · Decision Letter 0]

30 Oct 2020

Dear Dr. Hutson,

Thank you for submitting your manuscript "Can you teach a new mouse old tricks? Humanized mice as a model for Variola virus". Your manuscript was fully evaluated at the editorial level and by three independent peer reviewers. As you will see, the reviewers appreciated the importance of the topic but identified numerous aspects of the manuscript that must be improved before final consideration for publication. Most of the comments of Reviewers 1 and 3 can be dealt with by modifications of the text except for the queries regarding replication of the virus infectivity assays and negative control staining of anti-VARV antibodies. Reviewer 2 provided a much more critical response, which I largely agree with. Although the reviewer is correct in the assessment that replication of the results would greatly improve the study, I have decided that this will not be a condition of acceptability. Nevertheless, changes in the presentation of the data as outlined are needed to evaluate the significance of the work. I confirmed that Hu-BLT mice could not be found in a search of the JAX website and this mouse strain needs to be fully described

Sincerely,

Bernard Moss, M.D., Ph.D.

Guest Editor

PLOS Pathogens

Erle Robertson

Section Editor

PLOS Pathogens

Kasturi Haldar

Editor-in-Chief

PLOS Pathogens

orcid.org/0000-0001-5065-158X

Michael Malim

Editor-in-Chief

PLOS Pathogens

orcid.org/0000-0002-7699-2064

Reviewer Comments (if any, and for reference):

Reviewer's Responses to Questions

**Part I - Summary**

Reviewer #1: Background

Smallpox was a human disease caused by variola virus (VARV) and small animal models of smallpox are not available. Due to fear of bioterrorism or re-emergence of smallpox, the WHA authorised research with VARV to develop drugs, diagnostics and a safer vaccine. Testing of such medical countermeasures (MCMs) has been hindered by the lack of a suitable small animal model. Nonetheless, in 2018 the US FDA approved use of an anti-smallpox drug (tecovirimat) using data derived from animal models with surrogate orthopoxviruses. Given that orthopoxviruses can become resistant to tecovirimat, another drug with a different mechanism of action is needed. Here, the authors have explored 3 different humanized mice as possible models in which to test promising new antivirals.

Summary

The study describes the use of humanized mice for infection with VARV and shows that BLT and CD34+ humanized mice support VARV replication and systemic spread to develop disease with some similarity to human smallpox. The authors provide evidence of VARV infection in different tissues by histological and immunohistological analyses and show the presence of orthopoxvirus particles in infected cells by electron microscopy and the presence of VARV DNA by PCR.

Reviewer #2: The manuscript by Hutson presents a new mouse model for variola virus infection using mice that have a humanized hematopoietic system. The results are striking and interesting in that a systemic disease with some features similar to smallpox is shown in some individual mice, especially for the HuCd34+ and HuBLT mice. The study presents data that one would expect to see in the characterization of a new model and there is little doubt that the infection and pathogenesis are unique in the mice engrafted with human hematopoietic cells. However the majority of the data are derived from a single experiment that has a small number of mice per group (4 or 3 only). There is also a focus on presentation of histological findings that are qualitative by nature, which is helpful, but also tends to lend the feel of a case study to the paper, rather than a more rigorously quantitative approach. Even the virus loads are presented on a per-animal basis instead of being compiled in a way that makes basic statistical parameters easy to assess mean/median and a representation of error or distribution. If this model is to be used to test new drugs for smallpox, it is necessary that there is a better understanding of the range of all parameters that might be measured across a group of mice so that the study can be adequately powered. With the current data and especially the way it is presented, this is difficult to glean. On the one hand it is important to acknowledge that the work that is presented here, which requires BSL4 containment, is extremely difficult to do. However on the other, the intent is for other studies to follow the path that is set by this publication, so it is imperative that this not simply leave an impression of what is happening, but provide some firm quantitative information. In balancing these considerations, more data and at least one independent replicate for each strain/dose that the authors would advocate for future use would help greatly to cement the value of the study.

Reviewer #3: The authors describe new mouse models for smallpox. Three humanized mice (hu-PBMC, hu-CD34+ and hu-BLT) were infected with variola virus (VARV) and shown to be susceptible to infection. The infection was lethal for hu-CD34+ and hu-BT mice, whereas the virus replicated in hu-PBMC mice but caused a mild disease with no mortalities. A description of the pathology and virus replication is provided.

The finding of mouse models of smallpox that allow the study of virus replication and pathological consequences of the infection is of high interest in the field. These models will be useful to evaluate candidate anti-viral drugs, although the study of vaccines and immune response will be limited (human IgG and IgM were not detected in the initial studies). Also, they bring fundamental and interesting questions on the human susceptibility to VARV and the factors that confer resistance to infection in mice. The manuscript is mainly descriptive (necessary in the first report of these models) and the discussion could be expanded, even if some hypotheses may need additional experimentation.

**Part II – Major Issues: Key Experiments Required for Acceptance**

Reviewer #1: This reviewer is not requesting further experimentation, unless the virus infectivity titrations in figure 5 have only been conducted once without replicate assays.

Reviewer #2: As noted at the end of the general comments. There needs to be independent replication of the experiments and a total of more mice such that a better assessment of the frequency of all features of pathology in a group of each type of mouse and each dose can be presented.

Reviewer #3: One of the most interesting questions is to identify which components of the human immune system are required for VARV to spread in these mice. Further discussion on the component of the human immune system recreated in hu-PBMC mice (showing mild disease) as compared to hu-BLT and hu-CD34+ mice (showing severe disease) would be interesting and maybe informative.

VARV infects not only human immune cells but also mouse tissues, with very high virus titers in different organs. This is an interesting result since human cells may help virus spread to tissues that can support virus replication. More discussion on the infection of mouse tissues by VARV in this model and the existing evidence of the susceptibility of mouse cells to VARV would be helpful. This would emphasize the role of human immune cells in the establishment of a systemic infection in these mice.

**Part III – Minor Issues: Editorial and Data Presentation Modifications**

Reviewer #1: Could the authors please consider the following points?

1. Title. It is established that mice can be engineered to become susceptible to infection by different viruses to which normally they are resistant. This was shown first with polio virus and more recently there are other examples as mentioned in the introduction. Given this, the question posed in the first half of the title is already answered. So the title should be shortened by removal of the first part.

2. In several places the paper is drafted for a USA audience who may understand what a “select agent” is and what CDC and FDA are. For the benefit of the international readership of PLOS Pathogens could these terms be defined / explained more fully? e.g. US FDA, US CDC (there is also a CDC in Europe) and the nation from which terms such as “select agent” and “select agent regulations” derive be defined.

3. The abstract concludes that the model described may be useful for 3 things: testing MCMs for smallpox, identifying why VARV is a pathogen only of humans, and provides samples for understanding the human immune response to VARV infection. This reviewer agrees with the first point, but not the latter two and recommends these should be deleted.

4. Lines 54-55. It is not clear what “current recommendations” are referred to. Suggest deletion of “improves upon current recommendations by”.

5. Line 65-66. Reconstruction of horsepox virus. It may be misleading to suggest that horsepox virus (which is considered a strain of VACV) is closely related to VARV. Although all OPXVs are closely related, within the OPXV genus VACV and VARV are quite distinct and taterapox virus and camelpox virus are closest to VARV.

6. Line 96. define IACUC

7. Line 114. Specify if anesthesia was used. Similarly, line 129, define “5% inhalation anesthesia” – inhalation of what at 5%?

8. Line 155. “As” is missing from this sentence.

9. In general the figure legends i) do not provide enough detail of what was done or what is shown, and ii) contain too much interpretation of material presented, which should instead (mostly) be contained in the results narrative.

10. Fig. 1. The labeling and presentation of this figure needs improvement. Data presented in panels A and B are also included in C and D. So delete A and B. Please relabel figure with increased font size. In the legend define what high and low mean - this could easily be included on the figure itself. Change “survivorship” to “survival” (in text also) Day = day post infection.

11. Lines 178 – 184. If this animal was not infected correctly it should be removed from all data presented and the group size redefined.

12. Line 194. Please explain what “gamma-irradiated control hu-PBMC mouse” is. There is no mention of gamma-irradiation in the Fig. legend.

13. Fig. 2. Please re-write legend to describe clearly which animals these samples are from, what dose of virus was given, the day post infection the samples were harvested, and what VARV positive staining looks like. In the histological images please use arrows to point out the features to which the text refers. Ditto Fig. 3. In Fig. 3 arrows are referred to, but were not visible to this reviewer. Specify which panel(s) they are in.

14. Line 254 Define “select tissues”.

15. Fig. 5. Define “processed for viral culture”. Plaque assay, or something else? The number of replicate samples, the number of times the plaque assay (or other assay) was repeated need defining, and error bars should be added. A statistical analysis should be included to justify what is “significant”.

16. The legend to Fig. 5 defines a *. But no * is visible in the figure. Define “LOD”.

17. Line 282. Explain what “Gross findings were absent during necropsy” means.

18. Line 297. Consider changing “extremely” to “increasingly”.

19. Lines 318-321. Sentence repetition. This reviewer feels this sentence overstates what has been shown. The report shows that these mice can be infected with VARV and develop a disease with some similarity to human smallpox. The report has not shown that this model will be useful for studying VARV infection, or potentially features of the human immune response to infection. On the other hand, it is reasonable to propose that this model may be useful for evaluating the effectiveness of drugs against smallpox. Similar comment for last part of discussion.

20. In the discussion, the claim that the model has features similar to hemorrhagic smallpox, or ordinary smallpox, is not well founded base on data presented, are unnecessary and should be deleted. This will help reduce the discussion, which is rather too long.

21. Line 372 typos “pathogenby compareing”

22. Lines 371-83. Discussion on why VARV is a pathogen only of humans should include consideration of gene loss during VARV evolution, see Science, 2020 DOI: 10.1126/science.aaw8977. The focus on the O1L gene is more about why other related viruses do not infect man, and so might also be deleted to shorten the discussion.

Reviewer #2: 1) The paper is not written in a conventional way in which the results section has a narrative so that readers can easily follow what has been done. Rather this information has to be obtained by reading the Materials and Methods before the Results. This is unwieldy and will be more so if the manuscript was formatted with the Methods after the Discussion as per the journal style. Further, the data on the numbers of mice with different findings is put in supplementary tables, when they are key to understanding what is happening across each group of mice. These need to be in the main paper. In the second table the groups should be separated by dose as they are in the first table.

2) The issue of just how many mice were in each group is made more difficult to follow because the authors use “n” to indicate the number of mice with a particular finding and not the total group size as defined conventionally in statistics. Further statements about the frequency of signs of illness that aggregates this across the three different types of humanized mice is not helpful in understanding whether one model would be preferred over another. As noted above, key data like the virus titers need to be presented in a more conventional group-wise manner. Individual data can and should still be shown (as points), but with the titers from the same organ (and experimental group) plotted together. This would allow comparisons across organs or mouse types.

3) A key feature piece of information that is required to truly understand this model is the extent to which virus replication is supported in mouse cells versus human cells. This is accessible to some extent from the histopathology, immunohistochemistry and descriptions, that find in at least some cases that VARV is indeed replicating in mouse tissues (e.g. to cause hock lesions), but is not addressed systematically. For example it would be interesting to know how much of the virus titer in organs may have been contributed by the human cells in transit. Some assessment of this could be made by qPCR for human markers to see is virus levels correlated with human cell levels. Perhaps mice were perfused to limit this complication, but this is not mentioned. In any case, adding more data that explores this issue would add more broad scientific interest to the paper.

4) The different humanized mice are not adequately introduced to the reader, nor are the results discussed in the context of the different biology of these models in a lot of depth. Indeed, the acronymns are not spelt out even in the materials and methods and a search of the Jax catalogue for “Hu-BLT” yeilds no results (the other strains could be found, but are given their full name, which is what is required in the manuscript). The discussion touches on differences in GVH (but again this is not explained) in these mouse strains, but there are also difference in the types of cells that develop in these mice. Given the role of macrophages/monocytes as an important cell for dissemination of ectromelia virus, the fact that these cells tend not to develop well in Hu-PBMC mice also be significant.

Reviewer #3: Line 89. It is mentioned that hu-BLT mice inoculated with low dose group succumbed to disease, but hu-CD34 mice also succumbed at low dose (Table S1 and Fig.1).

Negative control staining of anti-VARV antibodies against uninfected mouse tissues should be shown.

I could not see the legends to Supplementary Figures.

Table S1 should be included in the manuscript since it has relevant information.

PLOS authors have the option to publish the peer review history of their article (what does this mean?). If published, this will include your full peer review and any attached files.

Reviewer #1: No

Reviewer #2: No

Reviewer #3: No
---

## [Decision Letter · Decision Letter 1]

22 Mar 2021

Dear Dr. Hutson,

Thank you very much for submitting a revised manuscript "Can you teach a new mouse old tricks? Humanized mice as a model for Variola virus" for consideration at PLOS Pathogens. Although the reviewers felt that the manuscript was improved, some comments still need to be addressed. It would help and speed up the review process, if you responded in a point-by-point manner and more specifically indicated the changes made. We would also like to suggest a change in the title as humanized mice are an infection model for variola virus rather than a model for variola virus as stated. Also. rather than questioning whether you can teach a new mouse old tricks, you might change the title to the more affirmative "Teaching a new mouse old tricks: Humanized mice as an infection model for variola virus" as an example.

Based on the reviews, we are likely to accept this manuscript for publication, providing that you modify the manuscript according to the review recommendations.

[1] A letter containing a detailed (point-by-point) responses to all review comments, and a description of the changes you have made in the manuscript.

Sincerely,

Bernard Moss, M.D., Ph.D.

Guest Editor

PLOS Pathogens

Michael Malim

Section Editor

PLOS Pathogens

Kasturi Haldar

Editor-in-Chief

PLOS Pathogens

orcid.org/0000-0001-5065-158X

Michael Malim

Editor-in-Chief

PLOS Pathogens

orcid.org/0000-0002-7699-2064

Reviewer Comments (if any, and for reference):

Reviewer's Responses to Questions

**Part I - Summary**

Reviewer #1: (No Response)

Reviewer #2: In this revised manuscript, the authors have addressed most of the issues raised about the presentation of the data (noting some exceptions below), but not the substantive point made in my original review that the study remains limited by the very small group sizes and lack of replication, neither was this rebutted in any way. The new presentation of the virus titers further reinforce that variation is a substantial issue here that may make these models difficult to use. Interestingly the HuCD34+ mice seemed more promising in this regard with relatively consistent titers from most tissues rather than viral loads spanning many orders of magnitude as seen with the other types of mice.

Reviewer #3: The authors have addressed the concerns raised by the reviewers.

**Part II – Major Issues: Key Experiments Required for Acceptance**

Reviewer #1: (No Response)

Reviewer #2: There remains no replication of the experiments and no argument was made as to why this is not necessary.

Reviewer #3: None

**Part III – Minor Issues: Editorial and Data Presentation Modifications**

Reviewer #1: (No Response)

Reviewer #2: 1) The intranasal route of infection should be noted in the first sentence describing the experiment in the Results section and in at least the legend for the first figure. An outline of the experiment as a panel in the first figure showing a timeline with infection, starting numbers, when mice succumbed and sampling time would also make things clearest.

2) Statistical comparisons made in the text remain difficult to assess from the data shown in figures because often the groups being compared are separated on different graphs. E.g. a statistical comparison of mortality is made between doses for each models in the text, but the groups are plotted on two sets of axes in the figure, with the two doses on separate graphs.

3) The authors still use “n” to refer both to the total group size and the number of mice with a particular finding or are ambiguous in numerical descriptions. In one case this is “n=2/8”. In another sentence “involving up to 50% of liver in hu-CD34+ (n=3) and hu-BLT (n-=5)” – does this mean 50% of the livers in affected mice, or livers in 50% of the mice (but if the group sizes are 3 and 5, how does this make sense). The use of “n” should be exclusively for the group size. When reporting findings “1/5” as used in some places (or 1 of 5) should be used consistently. The reader should not be having to go to the tables to make sense of the text.

4) Do the virus loads correlate in any way to clinical findings? This should be explored and discussed.

5) Variation in the data should be discussed as this has an impact on the usability of these models to test drugs etc.

Reviewer #3: None

PLOS authors have the option to publish the peer review history of their article (what does this mean?). If published, this will include your full peer review and any attached files.

Reviewer #1: No

Reviewer #2: No

Reviewer #3: No

Figure Files:

Data Requirements:

Reproducibility:

References:

---

## [Editor Report · Decision Letter 2]

11 May 2021

Dear Dr. Hutson,

We are pleased to inform you that your manuscript 'Teaching a new mouse old tricks: Humanized mice as an infection model for Variola virus' has been provisionally accepted for publication in PLOS Pathogens.

Best regards,

Bernard Moss, M.D., Ph.D.

Guest Editor

PLOS Pathogens

Michael Malim

Section Editor

PLOS Pathogens

Kasturi Haldar

Editor-in-Chief

PLOS Pathogens

orcid.org/0000-0001-5065-158X

Michael Malim

Editor-in-Chief

PLOS Pathogens

orcid.org/0000-0002-7699-2064
---

## [Editor Report · Acceptance letter]

29 Jun 2021

Dear Dr. Hutson,

We are delighted to inform you that your manuscript, "Teaching a new mouse old tricks: Humanized mice as an infection model for Variola virus," has been formally accepted for publication in PLOS Pathogens.

Best regards,

Kasturi Haldar

Editor-in-Chief

PLOS Pathogens

orcid.org/0000-0001-5065-158X

Michael Malim

Editor-in-Chief

PLOS Pathogens

orcid.org/0000-0002-7699-2064